# Underwater Image Color Constancy Calculation with Optimized Deep Extreme Learning Machine Based on Improved Arithmetic Optimization Algorithm

**Junyi Yang** [1] , **Qichao Yu** [2], **Sheng Chen** [1] **and Donghe Yang** [3,*]

1    School of Mechanical Engineering, Hangzhou Dianzi University, Hangzhou 310018, China;
     junyiyang@hdu.edu.cn (J.Y.); chensh@hdu.edu.cn (S.C.)
2    School of Electronics and Information Engineering, Hangzhou Dianzi University, Hangzhou 310018, China;
     m13567577959@163.com
3    School of Computer Science and Technology, Zhejiang Sci-Tech University, Hangzhou 310018, China
*    Correspondence: yangdonghe@zstu.edu.cn

**Abstract:** To overcome the challenges posed by the underwater environment and restore the true colors of marine objects' surfaces, a novel underwater image illumination estimation model, termed the iterative chaotic improved arithmetic optimization algorithm for deep extreme learning machines (IAOA-DELM), is proposed. In this study, the gray edge framework is utilized to extract color features from underwater images, which are employed as input vectors. To address the issue of unstable prediction results caused by the random selection of parameters in DELM, the arithmetic optimization algorithm (AOA) is integrated, and the search segment mapping method is optimized by using hidden layer biases and input layer weights. Furthermore, an iterative chaotic mapping initialization strategy is incorporated to provide AOA with a better initial search proxy. The IAOA-DELM model computes illumination information based on the input color vectors. Experimental evaluations conducted on actual underwater images demonstrate that the proposed IAOA-DELM illumination correction model achieves an accuracy of 96.07%. When compared to the ORELM, ELM, RVFL, and BP models, the IAOA-DELM model exhibits improvements of 6.96%, 7.54%, 8.00%, and 8.89%, respectively, making it the most effective among the compared illumination correction models.

**Keywords:** illumination estimation; deep extreme learning machine; arithmetic optimization algorithm; iterative chaotic mapping

## 1. Introduction

In the field of marine observation, scientists utilize underwater robots to capture photographs for studying marine organisms and mineral resources. For instance, in the research of coral bleaching phenomena and underwater polymetallic nodule deposits, the analysis and recognition of the color characteristics of underwater targets are essential [1,2]. However, in complex underwater environments, various factors, such as lighting conditions, spectral absorption by water media, and backscattering of particles, hinder the direct reflection of the true colors of objects in images [3]. Even with the same light source, the resulting image colors can vary. This instability of colors in underwater images poses a challenge, as the technology to obtain the accurate surface colors of marine objects is not yet mature [4]. Therefore, studying color constancy in underwater images becomes an urgent problem to address.

Early methods for color constancy analysis in underwater images were based on the statistical analysis of pixel values. Among them, the white-patch algorithm [5,6] assumes that a white surface can adequately reflect the illuminant color of the scene by selecting the maximum value among the RGB color channels as the illuminant color for the image. However, this algorithm's estimation performance is not optimal when the overall scene

brightness is low. The gray-world algorithm [7,8] assumes that for color-rich images, the average pixel values of the three RGB color channels tend to be similar. However, this method's estimation performance is not ideal for images with limited color or a single color, making it challenging to apply in underwater environments. Li et al. [9] proposed a wavelet transform method based on the YUV color model, which significantly improves imaging quality. Yan et al. [10] proposed a new color constancy framework based on the relationship between the reflectance difference and the local normalized reflectance difference. Iqbal et al. [11] introduced the Laplacian transform to minimize artifacts and noise. With the advancement of underwater image processing techniques, Hassan et al. [12] constructed an underwater image processing method supported by Retinex theory and achieved better results. However, these aforementioned algorithms are manually designed and have certain limitations in their application, as they cannot effectively perform image illumination estimation under different lighting conditions and complex environments.

The application of machine learning techniques in color constancy research has provided a new direction for solving complex pattern recognition problems. In recent years, researchers have started to incorporate machine learning methods into color constancy analysis, particularly algorithms based on image feature learning. Different learning methods have been proposed for color constancy, such as Bayesian-based color constancy algorithms [13], backpropagation (BP) neural network-based color constancy algorithms [14], support vector regression (SVR)-based illuminance estimation algorithms [15], and extreme learning machine (ELM)-based illuminance estimation algorithms [16]. Furthermore, deep learning methods with more powerful learning capabilities have been added to the application of color constancy [17]. Deep learning color constancy methods based on convolutional neural networks (CNN) [18,19], transfer learning [20], fast Fourier transform-based color constancy methods [21], and contrastive learning for color Constancy [22] have been proposed. In deep learning algorithms, image features are determined during the training of the network, and the complex deep learning network models significantly increase the computational burden. To combine the powerful computational capabilities of deep learning algorithms with the efficient learning ability of ELM, researchers have extensively investigated the integration of ELM into general deep learning frameworks. Deng et al. [23] presented the DELM by combining ELM with the idea of autoencoders, called the Extreme Learning Machine Autoencoder (ELM-AE). DELM exhibits strong nonlinear modeling and generalization abilities, which can be further expanded by using larger-scale training data and deeper network structures, showing great potential in the application of underwater image color constancy computation. Some studies have found that applying a swarm intelligence algorithm to find the optimal parameters of DELM is very helpful to its performance [24–26].

Motivated by the aforementioned research, this study proposes an iterative chaotic-based arithmetic optimization algorithm (IAOA) to optimize the light estimation model of DELM, referred to as IAOA-DELM. For underwater images captured under unknown illuminations, the IAOA-DELM approach first extracts the color features of the images using the gray edge framework [27]. Then, based on IAOA-DELM, it estimates the light source of the images. Finally, the estimated light source is applied to correct the underwater images to standard illuminant values using the Von Kries diagonal model [28]. This correction ensures that the underwater images collected under different color temperature lighting conditions exhibit color-accurate results. The innovation of this paper is primarily demonstrated in the following aspects:

(1)　Constructing the basic model of DELM to compute scene illumination information from the color features of underwater images.

(2)　To address the stability and generalization issues caused by the initial parameters in the orthogonal matrix, AOA is employed to optimize the input layer weights and thresholds of ELM-AE in the DELM structure. The search and development stages of AOA are combined with the nonlinear feature mapping stage of ELM-AE.

(3)     AOA is applied to select the hidden layer nodes' number and adaptively search for the optimization of effective activation nodes. It simultaneously optimizes hidden layer biases, input weights, and hidden layer nodes' numbers, obtaining an underwater image illumination estimation model with good predictive performance and stability.

(4)     The overall initial search agents of AOA are generated using iterative chaos mapping to improve the initialization strategy of AOA and obtain IAOA. In the initialization strategy, without prior knowledge, IAOA enhances the initial population's quality, thereby improving the algorithm's operation speed and accuracy.

The other chapters can be divided into the following sections: Section 2 provides an introduction to AOA and DELM. Section 3 describes the image dataset built and presents the proposed color constancy model (IAOA-DELM). Section 4 describes the analysis of the experimental results. Finally, Section 5 is the summary of the article.

## 2. Theoretical Basis

### 2.1. Arithmetic Optimization Algorithm

AOA is a meta-heuristic optimization algorithm based on the distribution behavior of arithmetic operators (addition, subtraction, multiplication, and division) [28]. It is characterized by the algorithm's fast convergence speed and high accuracy.

The AOA algorithm determines which search phase to enter according to the relationship between $r_1$ and the value of the mathematical optimizer acceleration function (A). $r_1 \in [0, 1]$. When $r_1 > A$ for global exploration; when $r_1 < A$, AOA is developed locally. A is shown as follows:

$$A(C_{iter}) = Min + C_{iter} \times \left( \frac{Max - Min}{M_{iter}} \right) \tag{1}$$

where $A(C_{iter})$ represents the value of the current iteration. $C_{iter}$ represents the current iteration. $M_{iter}$ represents the total number of iterations. Min and Max represent the maximum and minimum values of the acceleration function, respectively.

The AOA algorithm uses multiplication and division strategies for global search. $r_2$ is a random number between 0 and 1. When $r_2 < 0.5$, AOA enters the division search stage. When $r_2 > 0.5$, AOA enters the multiplication search stage. The position update formula of the multiplication and division strategy is as follows:

$$X_{i,j}(C_{iter} + 1) = \begin{cases} best(X_j)/(MOP + \varepsilon) \times ((UB_j - LB_j) \times \mu + LB_j), r_2 < 0.5 \\ best(X_j) \times MOP \times ((UB_j - LB_j) \times \mu + LB_j), r_2 > 0.5 \end{cases} \tag{2}$$

where, $X_{i,j}(C_{iter} + 1)$ represents the jth position of the ith solution in the next iteration, and $best(X_j)$ represents the best obtained solution's jth position so far. $\varepsilon$ is a very small integer, $UB_j$ and $LB_j$ represent the upper and lower bounds at the jth position. $\mu$ is a control parameter that adjusts the search process, which is fixed at 0.5 in this research. The mathematical optimizer probability (P) is shown as follows:

$$P(C_{iter}) = 1 - \frac{C_{iter}^{\alpha}}{M_{iter}^{\alpha}} \tag{3}$$

where, $\alpha$ defines the mining accuracy in the iterative operation process.

The AOA algorithm uses addition and subtraction strategies for local development. $r_3$ is a random number between 0 and 1. When $r_3 > 0.5$, AOA enters the phase of addition search. When $r_3 < 0.5$, AOA enters the subtraction search stage. The formula for the strategic position update for addition and subtraction is as follows:

$$X_{i,j}(C_{iter} + 1) = \begin{cases} best(X_j) - MOP \times ((UB_j - LB_j) \times \mu + LB_j), r_3 < 0.5 \\ best(X_j) + MOP \times ((UB_j - LB_j) \times \mu + LB_j), r_3 > 0.5 \end{cases} \tag{4}$$

### 2.2. Deep Extreme Learning Machine

Suppose there are N arbitrary samples $(X_j, t_j)$, $X_j = [x_{j1}, x_{j2}, ..., x_{jn}] \in R^n$ and $t_j = [o_{j1}, o_{j2}, ..., o_{jm}] \in R^m$, an ELM [29] with L hidden layer nodes can be shown as:

$$\sum_{i=1}^{L} \beta_i g(W_i \bullet X_j + b_i) = o_j, j = 1, 2, \cdots, N \tag{5}$$

where $g(x)$ is the activation function. $W_i = [w_{i1}, w_{i2}, ..., w_{in}]^T$ is the ith hidden layer neuron's input weight. $b_i$ is the ith hidden layer of layer neurons' bias. $\beta_j = [\beta_{j1}, \beta_{j2}, ..., \beta_{jm}]^T$ is the ith hidden layer neuron's output weight. $\bullet$ represents the inner product.

The output weight matrix is:

$$\hat{\beta} = H^+ T \tag{6}$$

where $H^+$ is the Moore-Penrose generalized matrix of the output matrix H.

DELM takes ELM as an autoencoder (ELM-AE) and randomly generates the parameter vector in ELM-AE. DELM employs the ELM-AE technique for training the first-layer network and obtaining the output of the first-layer hidden layer. This output serves as the input for training the second-layer network's output using a similar process, continuing until the final hidden layer. The output between the last hidden layer and the output layer is computed using the least squares method. The network of DELM is expressed in Figure 1.

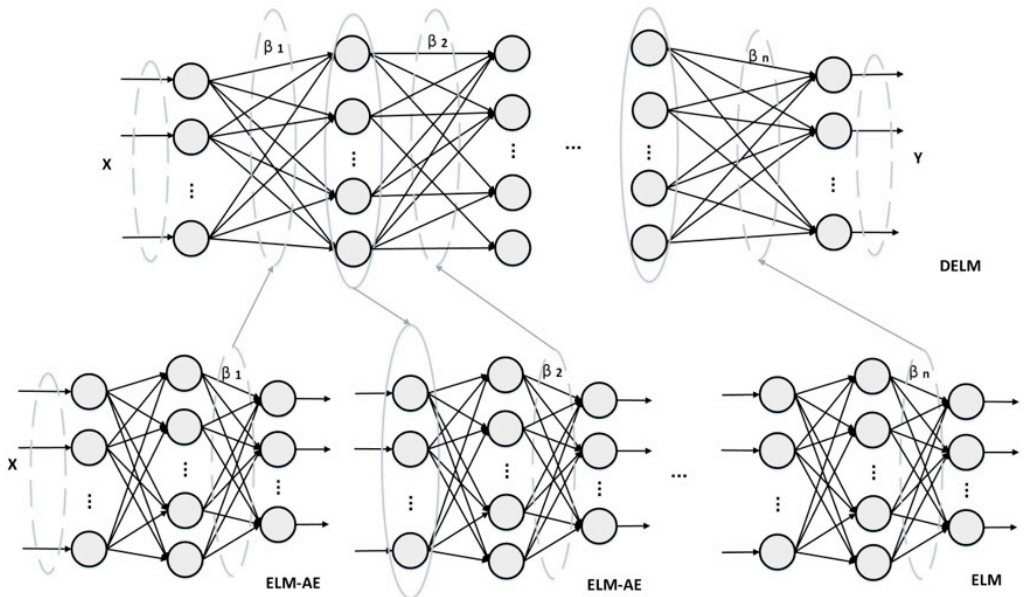

**Figure 1.** DELM Network structure.

DELM uses multiple ELM-AE superpositions for calculation and maps high-dimensional input data to low-dimensional space with low distortion. This method enables DELM to learn more abstract and useful features in the original data and achieve efficient feature representation.

### 3. Our Contribution

The selection of weights and biases for each hidden layer in ELM-AE is done randomly, and this randomness can have an influence on the training performance and stability of DELM. To address the instability issue in the selection of DELM parameters, the arithmetic optimization algorithm (AOA) was introduced. However, the AOA initialization strategy still suffers from the drawbacks of an uneven and unstable parameter distribution. To overcome this limitation, an iterative chaotic initialization strategy was incorporated to improve the AOA initialization. After optimizing the common parameters of DELM

using the improved AOA (IAOA), it is still necessary to conduct repeated experiments to determine the optimal values for the number of search agents, iteration count, hidden layer quantity, and number of neurons. This section explains the construction of the IAOA-DELM color constancy model and provides insights into the setup of the experimental scenarios and the construction of the dataset.

### 3.1. Search Agent Strategy of DELM Based on AOA

DELM employs an adaptive mechanism to activate effective neurons in the hidden layer based on the characteristics of the training data set. The number of hidden layer neuron nodes is a crucial parameter in the network structure, as it determines the number of relevant nodes in the input layer and the dimensionality of the effective feature parameters. Insufficient hidden layer nodes can impede DELM from accurately capturing the common features of the training set, while an excessive number of nodes may lead to an overemphasis on the specific features of the training set, thereby compromising its generalization ability. Traditional approaches for determining the optimal number of hidden layer nodes include conducting repeated experiments or relying on empirical knowledge to identify the ideal number. Alternatively, a fixed formula derived from statistical experience can be used to calculate the number of nodes in the hidden layer (m) based on the number of nodes in the input layer (n) and the number of nodes in the output layer (s), such as the formulas $m = \sqrt{ns}$ and $m = \log_2 n$. In the first method, researchers set the comparison interval and step size subjectively, which had high complexity and low accuracy. The second method only takes into account the number of nodes in the input and output layers as influential factors, neglecting other important considerations. It will miss the effective feature information of the data itself and produce large training errors. To address this issue, this study introduces a search agent strategy for DELM based on AOA optimization, aiming to determine the number of hidden layer neurons. The steps were as follows:

1.  To determine the maximum network structure, the number of hidden layers and the upper limit of the number of hidden layer nodes were set for DELM;
2.  The relevant parameters of AOA were initialized, and the n input nodes' number and the s hidden layer nodes' number were input into AOA as independent parameters for optimization;
3.  The fitness value of each individual was calculated to obtain the optimal parameter combination based on the search agent structure, and the node parameter results of the input and output layers were collected;
4.  According to the Ceil function, map the result to 0 or 1 (0 means freezing the node, 1 means activating the node), and calculate the number of optimal hidden layer nodes.

The search agent in the research is expressed in Figure 2:

Figure 2 depicts the search agent structure, which consists of four nodes in the input layer and five nodes in the hidden layer. The second and fourth nodes in the input layer are activated, while the first, fourth, and fifth nodes in the hidden layer are activated. Whether it is activated or not indicates that the network selects the effective input features of the second and fourth for training and connects the effective hidden layer nodes 1, 4, and 5. According to the connection relation between the effective nodes of the input layer and the hidden layer, the input layer weight ($W_{12}$, $W_{14}$, $W_{42}$, $W_{44}$, $W_{52}$, and $W_{54}$) and the hidden layer bias ($B_1$, $B_4$, and $B_5$) are selected, and the output parameter matrix is formed. Thus, the selection and learning of important features in the training set are completed. The optimization of input layer weight, hidden layer bias, and the number of hidden layer nodes are realized simultaneously. The input weight matrix β of ELM-AE was calculated according to formula (6). The output weight matrix of all hidden layers was obtained through training so as to obtain the AOA-DELM.

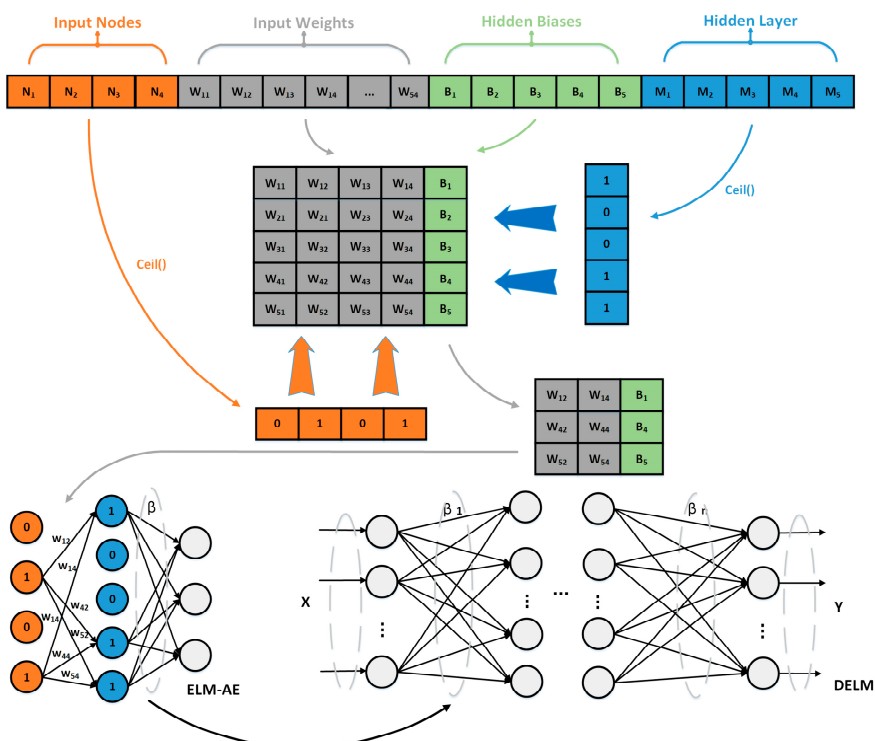

**Figure 2.** Search agent mapping flowchart.

### 3.2. Improved Arithmetic Optimization Algorithm Based on Iterative Chaotic Initialization (IAOA)

In the initial stage of the meta-heuristic optimization algorithm, the initial population is selected by means of random generation, which will lead to an uneven distribution of individuals in the population. Iterative chaotic mapping is characterized by strong chaos and pseudo-randomness. Compared with the initialization strategy of randomly generating the initial position, iterative chaotic mapping can make the population more evenly distributed in the search space [30]. If a population has n individuals, then the random initial population $X = \{x_1, \ldots, x_k, \ldots, x_n\}$, $k \in [1, n]$, and the mathematical expression is as follows:

$$x_{k+1} = \sin\left(\frac{a\pi}{x_k}\right) \tag{7}$$

where, $x_{k+1}$ is the value of introducing iterative chaotic mapping, $a \in (0,1)$. In this paper, a is 0.7.

AOA uses mathematical operators as an optimization means to select the population with the lowest loss function from all common populations (candidate schemes). However, the initial strategy of random distribution of AOA will lead to a large number of individuals in the population moving away from the optimal value, which limits the optimization efficiency of the AOA mechanism. If the population distribution is close enough to the optimal scenario, then the exploration and search phases of AOA will be efficient enough. In order to achieve this goal, an iterative chaos algorithm was introduced to form the initial population distribution of AOA into IAOA.

Suppose there are N search agents in IAOA initialization, each search agent vector has M dimensions, and each variable has the same upper boundary ub and the same lower boundary lb. Firstly, the first dimension value was randomly generated for the first search agent vector, $X_1$. The second dimension value $X_2$ was generated using the iterative chaos algorithm, $X_2$ was opposite to the distribution of $X_1$. Generate the first search proxy vector $X_2 = \{x_1, \ldots, x_k, \ldots, x_m\}$ based on the idea that $x_{k+1} = \sin(0.7\pi/x_k)$. Similarly, the population vector of the remaining N-1 search agents was obtained to form the initial search agent population matrix $X_2 = \{x_1, \ldots, x_N\}$ of IAOA. The strategy of boundary absorption was adopted in the subsequent iteration. The latitude value greater than the

upper boundary was set as ub, and the latitude value less than the upper boundary was set as lb.

Compared with the completely random generation of search proxy vectors by the original AOA algorithm, the initial population of IAOA is more widely distributed and more uniform in the search interval. The initial population, evenly distributed within the search interval, has a higher likelihood of capturing the correct eigenvalues, and it is more likely to explore and discover the optimal population.

### 3.3. Color Constancy Algorithm Flow of Underwater Image Based on IAOA-DELM

The IAOA-DELM color constancy algorithm for underwater images was constructed by combining the IAOA-DELM design idea with underwater image feature learning. The detailed steps and complete process were described as follows:

1. Underwater scene images were shot, and a gray edge frame was used to extract color features from the images as an input vector and constitute the input data set;
2. The number of DELM hidden layers, the number of iterations, and the number of search agents were input. A group of excellent initial populations for AOA was generated by using the iterative chaos algorithm;
3. The dataset was randomly divided into training and test sets using ten-fold cross-validation, where nine subsets were used for training and one subset was used for testing;
4. The training data set was input, the chromaticity feature vector was normalized, and the parameters were limited to search the effective interval. The training set was used as input for training, and the effective nodes of DELM were activated. The enhanced AOA algorithm was employed to optimize the input layer weights, hidden layer biases, and hidden layer nodes of DELM;
5. The fitness of the AOA search agent population was calculated and compared with the best fitness in the previous iteration to decide whether to update the population position;
6. The optimal parameters of IAOA-DELM were obtained after reaching the maximum number of iterations, and the input weight matrix $\beta$ of ELM-AE was calculated. The output weight matrix of DELM was obtained, and the IAOA-DELM illumination estimation model was constructed;
7. The IAOA-DELM illumination estimation model was used to calculate the illumination of the test set images. The color constancy of underwater images is realized by restoring the image to the standard light source based on the diagonal mapping matrix.

The implementation flow chart of the IAOA-DELM lighting correction model is shown in Figure 3.

### 3.4. Experimental Scene Construction

The experimental environment was a computer with the CPU model ARM R7-5800H @3.2GHz. All experiments in this paper were completed by MATLAB2017b software in a Windows 11 environment.

The underwater image acquisition system included a PC terminal, pool, camera, object, and light source. The camera was a Mercury II series high precision industrial camera produced by DaHeng Image Company, model MER2-1220-32U3C. The camera has a resolution of 4024 (H) × 3036 (V), a frame rate of 32.3 fps, and a pixel size of 1.85 μm × 1.85 μm. The optical lens has a resolution of 8 mega pixels and a focal length of 8.0 mm. Light source Seven common lighting sources recommended by the International Commission on Illumination (CIE) were used to provide lighting. The light source types are presented in Table 1.

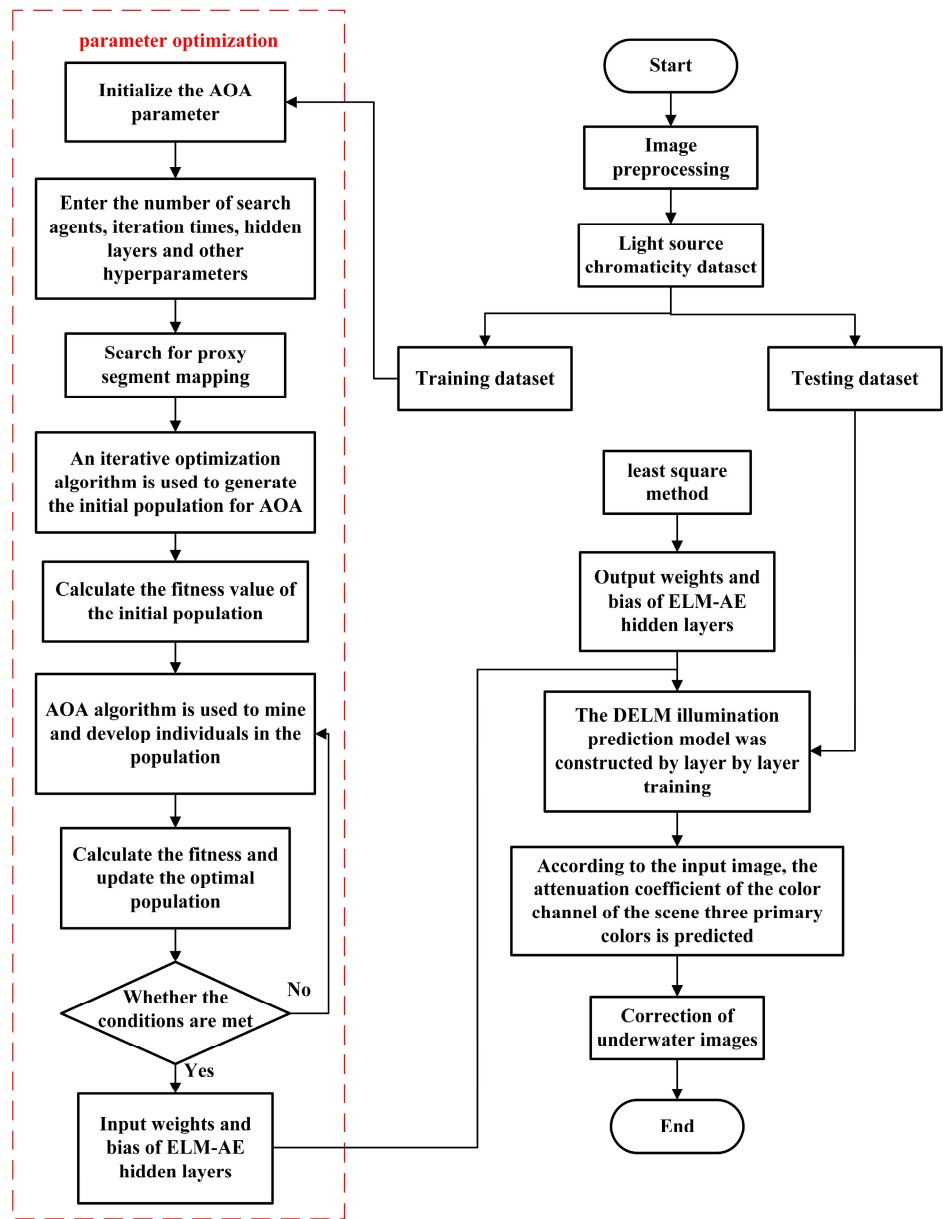

**Figure 3.** IAOA-DELM algorithm flowchart.

**Table 1.** The light source used in this article.

| Light Source Name | Color Temperature | Color Index | Type of Light Source |
|---|---|---|---|
| TL83 | 3000 K | 85 | European commercial fluorescence |
| U35 | 3500 K | 85 | American commercial fluorescence |
| TL84 | 4000 K | 85 | European commercial fluorescence |
| CWF | 4150 K | 62 | American commercial fluorescence |
| D50 | 5000 K | 95+ | Filtered tungsten lamp |
| D65 | 6500 K | 95+ | Filtered tungsten lamp |

In the layout of the pool environment, the filter was used to filter the impurities injected into the pool water. The method of reflection lighting was selected. The light source was installed on the same side of the camera to obtain the reflected image of the underwater object. The projection angle was guaranteed to remain unchanged after the replacement of the light source. To reduce the impact of bright spots, we painted all sides

except the front (where the camera looks in) black to reduce reflections from the air and water interfaces. In the shooting environment, the doors and windows were closed to ensure that there was only one experimental light source in the scene. The straight-line distance between the target and the camera was 1 m. The experimental environment is shown in Figure 4.

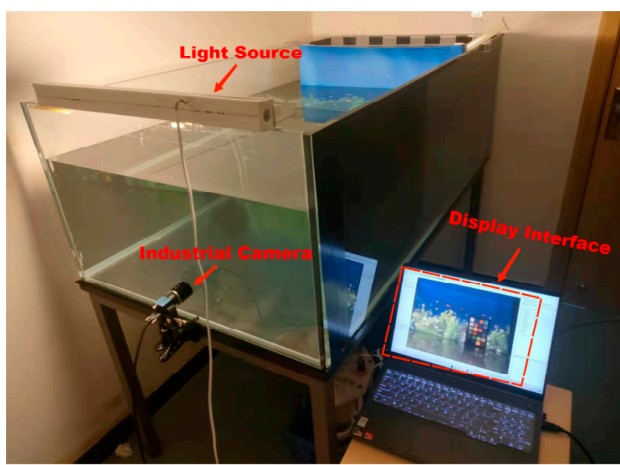

**Figure 4.** Experimental facility.

*3.5. Data Set Acquisition*

To enhance algorithm efficiency, it is essential to utilize an efficient and low-dimensional feature vector as the input for subsequent algorithms. Weijer et al. [31] introduced a gray-edge framework that consolidated various traditional unsupervised color constancy algorithms. The mathematical formula is presented as follows:

$$\left(\int \left|\frac{\partial^n f^\sigma(x)}{\partial x^n}\right|^p dx\right)^{1/p} = k e^{n,p,\sigma} = k e_i = (R_i, G_i, B_i) \tag{8}$$

where, $f^\sigma$ represents the convolution of image f and Gaussian filter $G^\sigma$.

By adjusting the parameters n, p and σ, different color constancy algorithms can be derived. The gray edge framework can be used as an image feature statistics tool to introduce higher-order derivative information of the image into the input features. In this research, the value range of n was selected as {0, 1, 2}, and the value range of p was selected as {1, 2, 3, ..., 10}. The value range of σ was {1, 3, 5, 7, 9}. There were 150 (3 × 10 × 5) cases in parameter combination i. When r and g were extracted, there were a total of 300-dimensional input feature vectors. In this data set, real illumination RGB information was added as a label by extracting color card information. Here, RGB information was converted to r and g chroma information to remove the influence of illumination intensity.

In this research, 300 underwater images were shot with six light sources of different color temperatures, as shown in Table 1.

The color features of the image dataset were extracted using the gray-edge framework. These extracted chromaticity features, along with the corresponding light source information obtained from the underwater color cards, formed the dataset. To evaluate the model, the dataset was divided into a training set and a test set using the ten-fold cross-validation method. The training set consisted of 270 samples, while the test set contained 30 samples. The training set was utilized to train the neural network and obtain the optimal model parameters. The test set was used to check the correction ability of the model and evaluate the performance of the algorithm.

*3.6. Evaluation Index*

Chroma error and angle error are important evaluation indexes of illumination correction. The true illumination of the image was set as $(r_c, g_c)$ and the illumination predicted by

the algorithm as $(r_u, g_u)$. The chroma error $(E_d)$ and angle error $(E_a)$ of the algorithm were shown in Equations (10) and (12). The smaller the angle error and chroma error, the better the effect of the algorithm. Chroma accuracy (CR) was used as the fitness of the search proxy population.

$$(r_i, g_i) = \left( \frac{R_i}{R_i + B_i + G_i}, \frac{G_i}{R_i + B_i + G_i} \right) \tag{9}$$

$$E_d = \sqrt{(r_c - r_u)^2 + (g_c - g_u)^2} \tag{10}$$

$$CR = 1 - E_d \tag{11}$$

$$E_a = \cos^{-1}\left( \frac{e_c \cdot e_u}{||e_c|| \, ||e_u||} \right) \times \frac{180°}{\pi} \tag{12}$$

where, $|| \cdot ||$ is the Euclidean norm of a vector.

The diagram of the IAOA-DELM light correction process is shown in Figure 5.

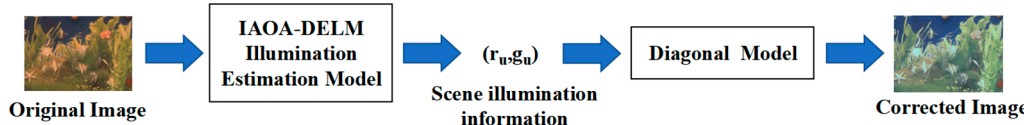

**Figure 5.** IAOA-DELM light correction process diagram.

## 4. Experimental Results and Analysis

### 4.1. Experimental Parameter Setting

#### 4.1.1. DELM Set Network Parameters

The aforementioned explanation provides a detailed account of how the IAOA-DELM algorithm accomplished the selection of hidden layer nodes in DELM using the search agent strategy. However, there are still two important hyperparameters that need to be set: the upper limit of hidden layer nodes and the number of hidden layers. There was no fixed correspondence between the super parameters of DELM and the regression prediction results, so the super parameters of DELM were determined by experimental comparison and cross-validation. For the purpose of fast training, this research discussed the chromaticity error of the algorithm in the six cases where the number of hidden layers of the IAOA-DELM algorithm was 2, 3, 4, 5, 6, and 7 to select the appropriate number of hidden layers. Based on the Kolmogorov theorem [32], the upper limit for the number of hidden layer nodes in this study was determined to be 600. The DELM algorithm employed an infinity regularization coefficient and utilized the sigmoid function as the activation function. To evaluate the performance of the algorithm on the same dataset, average accuracy and time cost were selected as the measurement indices. A total of fifty experiments were conducted using the ten-fold cross-validation method. The appropriate number of hidden layers was determined by analyzing the average accuracy, standard deviation, maximum accuracy, minimum accuracy, median accuracy, and time cost obtained from these fifty experiments. The experimental results are presented in Table 2.

Table 2 reveals that when the number of hidden layers is set to 2, the algorithm achieves optimal performance in terms of average accuracy, maximum accuracy, minimum accuracy, and median accuracy. Consequently, it is assumed that the number of hidden layers should be set to 2 in subsequent experiments. This demonstrates that an increasing number of hidden layers does not necessarily result in better performance. Excessive hidden layers can lead to a higher computational workload, a longer training time for the algorithm, and an overemphasis on the features of the training set, thereby hindering the accurate identification of image features by DELM.

**Table 2.** The influence of the number of different hidden layers in DELM on the experimental results (The best values are shown in bold).

| Number of Hidden Layers | Average Accuracy | Standard Deviation | Maximum Accuracy | Minimum Accuracy | Median Accuracy | Time Cost (s) |
|---|---|---|---|---|---|---|
| 1 | 94.47% | 0.0318 | **96.82%** | 88.91% | 94.47% | **0.1513** |
| 2 | **95.08%** | 0.0118 | 96.69% | **91.56%** | **95.31%** | 0.2624 |
| 3 | 89.71% | 0.0083 | 92.49% | 88.21% | 89.54% | 0.3699 |
| 4 | 89.32% | 0.0071 | 91.21% | 88.03% | 89.23% | 0.4801 |
| 5 | 89.21% | 0.0059 | 90.44% | 87.95% | 89.31% | 0.5796 |
| 6 | 89.27% | **0.0056** | 90.54% | 88.01% | 89.24% | 0.6649 |
| 7 | 88.68% | 0.0073 | 90.52% | 87.03% | 88.62% | 0.7578 |

The choice of activation function has a significant impact on the evaluation of light correction experiment results. To enhance the predictive performance of the model, this research analyzes chroma error $E_d$ using the ten-fold cross validation method for different activation functions. The table presents $E_d$ of the illumination estimation model on the dataset. From Table 3, it is evident that the Sigmoid activation function yields the smallest average $E_d$ among the ten-fold experimental results, indicating superior prediction performance. Hence, this paper adopts the Sigmoid activation function for the DELM model.

**Table 3.** $E_d$ of different activation functions.

| Activation Function | Sigmoid | Sine | Hardlim | Sign | Tribas |
|---|---|---|---|---|---|
| $E_d$ | 0.0387 | 0.0465 | 0.0415 | 0.0404 | 0.0541 |

4.1.2. Parameter Selection of IAOA

The value of the mathematical optimizer acceleration function A in AOA is influenced by the sensitive parameter $\alpha$, which plays a crucial role in determining the accuracy of the mining process during iterations. Algorithm iteration times and AOA population size also have a decisive influence on the optimization results of the IAOA algorithm. In order to obtain the optimal parameter combination of sensitive parameter alpha, population size, and iteration times, the average chrominance accuracy of the algorithm was taken as the measurement index. The research carried out an in-depth analysis of the influence of changes in these three parameters on the experimental results. Among them, the sensitive parameter alpha has three values of 0.25, 0.5, or 0.75; the population size has six values of 10, 20, 30, 40, 50, or 60; and the number of iterations has six values of 20, 40, 60, 80, 100, or 120, which has 108 different parameter combinations. The average chrominance accuracy under each parameter combination was obtained by means of 50 experiments. The experimental results are presented in Figures 6–8. The x-axis represents the population size, while the y-axis represents the number of iterations. The bubble size represents the average chrominance prediction accuracy.

In Figures 6–8, the highest average chrominance accuracy is achieved when alpha is set to 0.5 with a population size of 30 and 100 iterations. Due to the comprehensive consideration of computational complexity, operation efficiency, and accuracy, this parameter combination was selected as the preset parameter in subsequent experiments.

*4.2. Compare the Experimental Results of the Group*

4.2.1. Comparison Group Algorithm Parameter Selection

To assess the performance of the proposed IAOA-DELM illumination correction model, a comparison group consisting of eight neural network models was constructed, including the AOA-DELM, GWO-DELM, WOA-DELM, HHO-DELM, ORELM, ELM, RVFL, and BP models. The experimental results involved comparing the proposed algorithm with a comparison group algorithm and analyzing the impact of algorithm parameters. Table 4 provides the parameters for each algorithm.

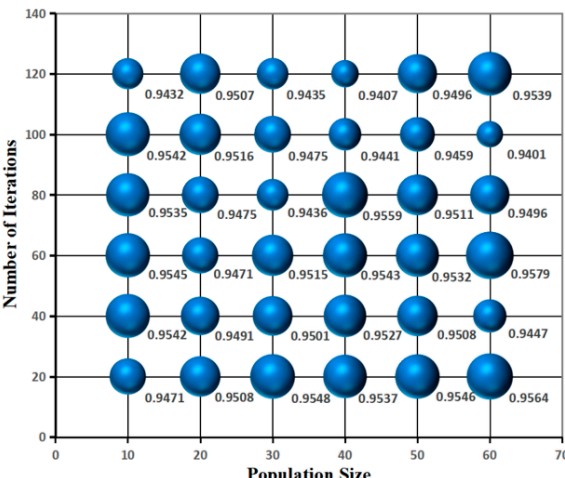

**Figure 6.** The average chrominance accuracy under different population sizes and number of iterations when alpha = 0.25.

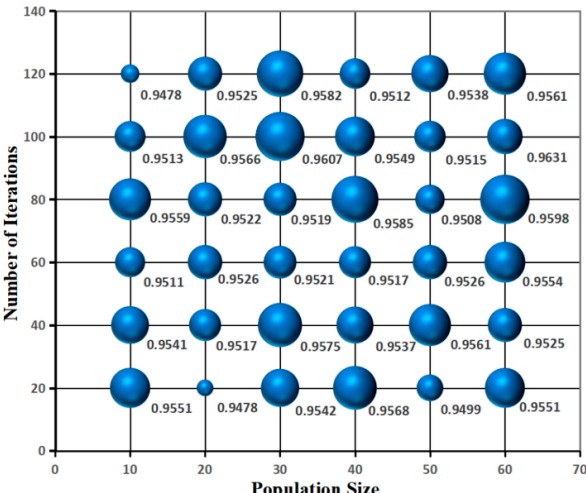

**Figure 7.** The average chrominance accuracy under different population sizes and number of iterations when alpha = 0.5.

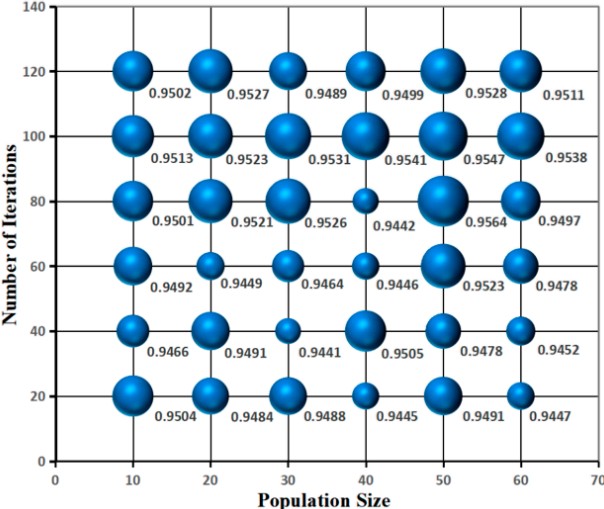

**Figure 8.** The average chrominance accuracy under different population sizes and number of iterations when alpha = 0.25.

**Table 4.** Parameter setting of extreme learning machine and its optimization algorithm.

| Algorithm | Parameter | Value |
|---|---|---|
| AOA-DELM | Number of populations | 30 |
| | $Iter_{Max}$ | 100 |
| | $P_{Max}$ | 1 |
| | $P_{Min}$ | 0.2 |
| | Alpha | 5 |
| GWO-DELM | Number of populations | 30 |
| | $Iter_{Max}$ | 100 |
| WOA-DELM | Number of populations | 30 |
| | $Iter_{Max}$ | 100 |
| HHO-DELM | Number of populations | 30 |
| | $Iter_{Max}$ | 100 |
| ORELM [33] | C | $2^{30}$ |
| ELM | NumberofHiddenNeurons | 600 |
| RVFL [34] | Option.mode | 2 |
| | Option.Scale | 1 |
| | Option.ActivationFunction | "sig" |
| | Option.Scalemode | 3 |
| BP | Net.trainParam.epochs | 100 |
| | Net.trainParam.Ir | 0.1 |
| | Net.trainParam.goal | 0.001 |

Abbreviations: AOA, Arithmetic Optimization Algorithm; GWO, Grey Wolf Optimization Algorithm; WOA, Whale Optimization Algorithm; HHO, Harris Hawks Optimization Algorithm; ORELM, Extreme Learning Machine with Outlier Robustness; ELM, Extreme Learning Machine; RVFL, Random Vector Function Link Network; BP, Back Propagation.

### 4.2.2. Comparison of Chroma Estimates

In order to further show the difference in the degree of fitting accuracy among algorithms, each group of algorithms was tested 50 times to obtain average chrominance accuracy. A histogram was drawn on the basis of the statistical table of experimental results. The abscissa is each algorithm, and the ordinate is the average chrominance accuracy under 50 experiments. The experimental results are shown in Figure 9.

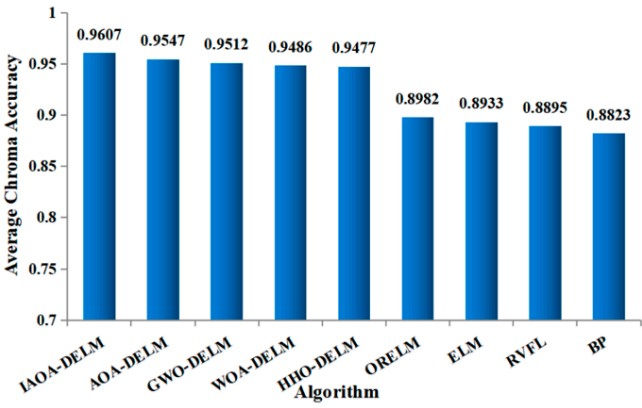

**Figure 9.** Comparison of average accuracy results between IAOA-DELM algorithm and comparison group algorithm.

Data in Figure 9 shows that the average prediction accuracy of the IAOA-DELM model is 96.07%, which is the maximum among all comparison models. Compared with AOA-DELM, GWO-DELM, WOA-DELM, and HHO-DELM, the average prediction accuracy of the IAOA-DELM model is increased by 0.63%, 0.99%, 1.27%, and 1.37%, respectively. This indicates that AOA has better searching ability, and improved AOA has higher searching

accuracy than AOA. Compared with ORELM, ELM, RVFL, and BP, the average prediction accuracy of the IAOA-DELM model is improved by 6.96%, 7.54%, 8.00%, and 8.89%, respectively. The results indicate a significant improvement in accuracy with the proposed algorithm compared to the traditional single hidden layer network algorithm. Table 5 shows the running time cost of optimizing DELM by various swarm intelligent algorithms. The research findings demonstrate that the IAOA-DELM model proposed in this study achieves high accuracy while consuming less time. Although the IAOA-DELM incurs slightly higher time costs compared to the AOA-DELM, it delivers superior precision. Considering the significance of accuracy improvement in specific applications, compromising on time costs is justifiable when sufficient time is available to achieve enhanced precision.

**Table 5.** The running time cost of each algorithm.

|               | IAOA-DELM | AOA-DELM | GWO-DELM | WOA-DELM | HHO-DELM |
| --- | --- | --- | --- | --- | --- |
| Time cost (s) | 18.31 | 17.97 | 28.46 | 29.73 | 41.45 |

4.2.3. Stability Analysis

The average angle error was recorded in the ten-fold cross verification, and the experimental results are shown in Figure 10. The more compact the box in the figure, the smaller the gap between the upper and lower quartiles, the more concentrated the distribution of experimental results, and the better the stability of the algorithm. The red line in the figure represents the median angle error. The lower the position of the red line, the smaller the overall prediction error of the algorithm. According to the angle error distribution analysis in Figure 10, the IAOA-DELM algorithm proposed in this research has the smallest prediction error. Its median average angle error is the lowest among all comparison groups. Its stability is higher than ORELM, ELM, RVFL, and BP.

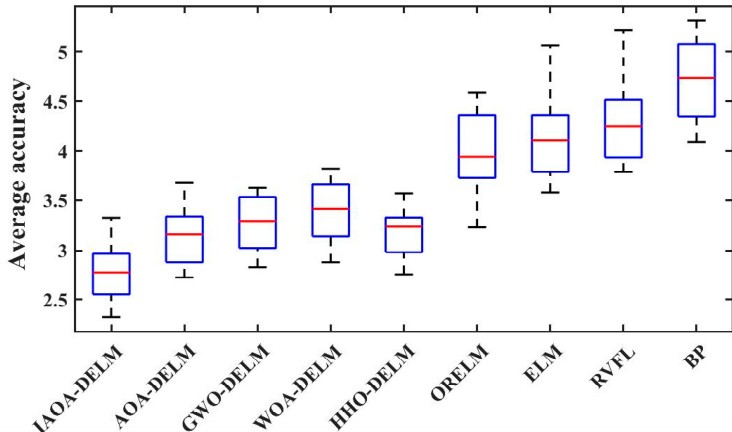

**Figure 10.** Algorithm angle error stability analysis box diagram.

4.2.4. Comparison of Image Correction Effect

To validate the algorithm's correction ability, a series of images were captured in both air and water environments. One of the underwater images taken by the TL83 light source was imported into each comparison algorithm for illumination estimation and used the diagonal model to restore the image. The recovery results of each algorithm are shown in Figure 11. The corrected image effect of the IAOA-DELM algorithm proposed in this research is the closest to the image taken in the air by the standard light source D50. AOA-DELM, GWO-DELM, WOA-DELM, and HHO-DELM are not as accurate as the algorithms proposed in this research, but they have certain correction effects in the performance of image restoration, which indicates that the DELM model has a good illumination prediction effect. The illuminance estimation abilities of ORELM, ELM, RVFL, and BP are weak, and

the suppression effect on the R channel is weak in image restoration. The corrected image retains the low color temperature to varying degrees.

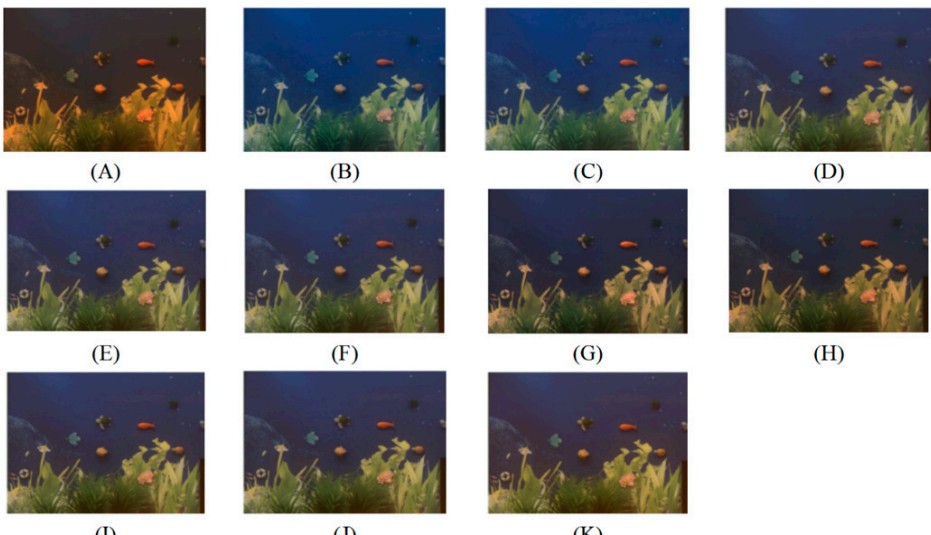

**Figure 11.** Correction results of different illumination correction models (**A**) Underwater image to be corrected under TL83 light source (**B**) Standard image in air under the same scene under D50 light source (**C**) IAOA-DELM (**D**) AOA-DELM (**E**) GWO-DELM (**F**) WOA-DELM (**G**) HHO-DELM (**H**) ORELM (**I**) ELM (**J**) RVFL (**K**) BP.

After confirming the application advantages of IAOA-DELM compared with the other eight artificial neural network models, this research selected some classical color constancy algorithms and advanced color constancy algorithms to form a comparison group for comparison with the algorithm proposed in this research. The data set images were imported into each algorithm to obtain the angle errors of each group of algorithms, and the mean, median, upper, and lower quartiles of the angle errors of each group were calculated. The experimental results are presented in Table 6, while Figure 12 displays the image restoration outcomes.

**Table 6.** Correction results of different illumination correction models (Minimum values are shown in bold).

| Algorithm | Mean | Med | Best 25% | Worst 25% |
|---|---|---|---|---|
| Interactive WB Method [35] | 3.63 | 3.46 | 0.94 | 6.94 |
| Data-Driven WB Method [36] | 3.49 | **3.04** | 0.87 | 5.86 |
| WB color augmenter [37] | 3.18 | 3.53 | **0.79** | 6.37 |
| Grey-World [7] | 7.81 | 7.34 | 1.62 | 9.78 |
| White-Patch [5] | 6.14 | 5.76 | 1.03 | 9.31 |
| Shades of Grey [38] | 5.12 | 4.86 | 1.12 | 9.76 |
| IAOA-DELM (ours) | **2.81** | 3.12 | 0.91 | **4.32** |

The analysis of Table 6 reveals that the IAOA-DELM algorithm exhibits the smallest angle error in both the mean and lower quartile, the deviation of IAOA-DELM is small, and the stability is strong. Figure 12 clearly illustrates that the IAOA-DELM algorithm outperforms several classical and advanced color constancy algorithms in terms of performance.

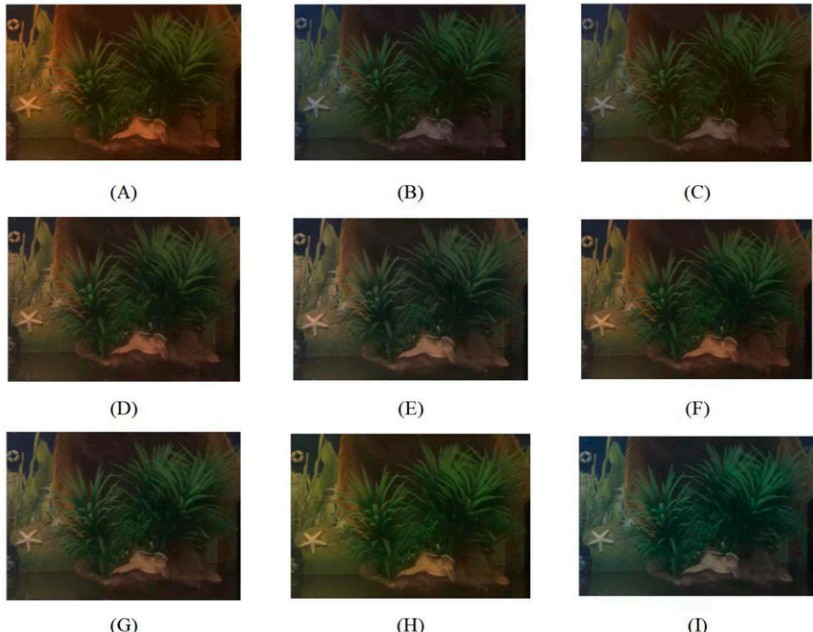

**Figure 12.** Comparison with other classical and advanced image correction algorithms (**A**) Underwater image to be corrected under TL83 light source (**B**) Standard image in air under the same scene under D50 light source (**C**) IAOA-DELM (**D**) Interactive WB Method (**E**) Data-Driven WB Method (**F**) WB color augmenter (**G**) Grey-World (**H**) Max-RGB (**I**) Shades of Grey.

## 5. Conclusions

This research proposes an iterative, chaotic, improved arithmetic optimization algorithm for deep extreme learning machines for the estimation of image illumination. Based on the experimental results, the following conclusions can be drawn:

(1) By optimizing the initial search agents of the AOA algorithm using the iterative chaotic map, this study achieves a more uniform initialization of the population. Comparing the results of computational accuracy in terms of time cost (shown in Table 5) and average accuracy (depicted in Figure 9), it is evident that this method significantly improves the accuracy of color constancy computation at a minimal time cost;

(2) By mapping the search agent fragments, the improved AOA algorithm optimizes the input weights and hidden biases of DELM. The stability analysis boxplot of algorithmic angular error, as presented in Figure 10, demonstrates that the improved AOA-DELM model exhibits a smaller angular error and good stability in color constancy;

(3) Comparison of the correction results with selected classical color constancy algorithms and advanced color constancy algorithms, as illustrated in Figure 12, verifies that the underwater image corrected by the improved AOA-DELM model achieves the best visual effect, resembling images with no color error in the air.

In conclusion, this model can improve the efficiency of illumination correction of underwater images while maintaining a small-time cost, which can be considered an efficient underwater image color constancy model.

**Author Contributions:** Conceptualization, J.Y. and Q.Y.; methodology, J.Y.; software, Q.Y.; validation, J.Y., Q.Y. and S.C.; formal analysis, S.C.; investigation, J.Y.; resources, J.Y.; data curation, Q.Y.; writing—original draft preparation, Q.Y.; writing—review and editing, J.Y.; visualization, J.Y.; supervision, S.C.; project administration, D.Y.; funding acquisition, J.Y. All authors have read and agreed to the published version of the manuscript.

**Funding:** This research is funded by the National Key Research and Development Program of China (2022YFC2803903).

**Data Availability Statement:** The data that support the findings of this study are available on request from the corresponding author.

**Acknowledgments:** We are very grateful to Zhiyu Zhou from Zhejiang Sci-Tech University and Chao Wang from Hangzhou Dianzi University for their guidance on my research methods, and the Marine Center of Hangzhou Dianzi University for providing the experimental site and equipment.

**Conflicts of Interest:** The authors declare no conflict of interest.

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
