# Peer review of "Underwater Image Color Constancy Calculation with Optimized Deep Extreme Learning Machine Based on Improved Arithmetic Optimization Algorithm"

_electronics, doi:10.3390/electronics12143174_

Round 1

Reviewer 1 Report

An advanced model of color correction is developed. It is based on deep extreme learning machine approach. The paper contains all necessary parts: introduction, theoretical consideration, method and database description, results. The language and logic are fine and easy to understand. There are the following remarks.

Formulas look not very elegant. In notations like 'C_iter' it is better to put 'iter' as subscript. Also some subscripts are then drawn as regular text. Defining values by several letters (like 'MOP' or 'MOA') is not very good. Using 'times' sign as in (1) and (2) and 'div' sign as in (2) is not good, since these signs are normally reserved for other purposes. It is commonly accepted that multiplication is usually not defined at all, and division is defined by '/' sign or stacked formulas. The authors should redesign their notations to be more compliant with commonly accepted in mathematics.

Line 118: 'size' is not correct. Should be 'value' or 'level'.

Reviewer 2 Report

A method of color correction based on deep extreme learning is presented. The work is theoretically interesting and has good practical applications. It can be published in the journal.

Reviewer 3 Report

The paper presents an interesting approach for Underwater image color constancy calculation through deep learning. The author needs to resolve the following issues.

 1.         The article uses deep neural networks for calculating color constancy in underwater images, but there is no further discussion of the architecture or model. More model specifics (layers, activations, and optimizations) would improve technical rigor.

2.         Although the study demonstrates success in calculating color constancy in underwater images, a better understanding of scalability and generalizability would benefit from further information on datasets (characteristics, size).

3.         Although the paper emphasizes the advantages of IAOA-DELM learning model, it does not address the difficulties (computational cost) that might limit its viability.

4.         To provide a visual representation and improve understanding of the discussed details, I advise adding an additional image to the paper. This image can clearly illustrate the concepts and information presented, aiding the reader's comprehension.

5.         Overall, the paper presents a promising method for deep learning-based underwater image colour constancy calculation. By addressing the aforementioned issues, the technical content would be strengthened and readers and researchers in the field would gain more knowledge.
